# Antimycobacterial Activities of Hydroxamic Acids and Their Iron(II/III), Nickel(II), Copper(II) and Zinc(II) Complexes

**DOI:** 10.3390/microorganisms11102611

**Published:** 2023-10-23

**Authors:** Dong Yang, Yanfang Zhang, Ibrahima Sory Sow, Hongping Liang, Naïma El Manssouri, Michel Gelbcke, Lina Dong, Guangxin Chen, François Dufrasne, Véronique Fontaine, Rongshan Li

**Affiliations:** 1Clinical Laboratory, Shanxi Provincial People’s Hospital, Affiliated of Shanxi Medical University, Taiyuan 030001, China; yangdcmu@163.com (D.Y.);; 2Microbiology, Bioorganic and Macromolecular Chemistry Unit, Faculty of Pharmacy, Université Libre de Bruxelles (ULB), Boulevard du Triomphe, 1050 Brussels, Belgium; ibrahima.sory.sow@ulb.be (I.S.S.); veronique.fontaine@ulb.be (V.F.); 3Core Laboratory, Shanxi Provincial People’s Hospital (Fifth Hospital) of Shanxi Medical University, Taiyuan 030012, China; 4Institutes of Biomedical Sciences, Shanxi University, Taiyuan 030006, China; 5Department of Nephrology, Shanxi Kidney Disease Institute, The Affiliated People’s Hospital of Shanxi Medical University, Shanxi Provincial People’s Hospital, Taiyuan 030001, China

**Keywords:** hydroxamic acids, antibacterial, anti-biofilm, *Mycobacterium tuberculosis*

## Abstract

Hydroxamic acid (HA) derivatives display antibacterial and antifungal activities. HA with various numbers of carbon atoms (C_2_, C_6_, C_8_, C_10_, C_12_ and C_17_), complexed with different metal ions, including Fe(II/III), Ni(II), Cu(II) and Zn(II), were evaluated for their antimycobacterial activities and their anti-biofilm activities. Some derivatives showed antimycobacterial activities, especially in biofilm growth conditions. For example, 20–100 µM of HA10Fe2, HA10FeCl, HA10Fe3, HA10Ni2 or HA10Cu2 inhibited *Mycobacterium tuberculosis*, *Mycobacterium bovis* BCG and *Mycobacterium marinum* biofilm development. HA10Fe2, HA12Fe2 and HA12FeCl could even attack pre-formed *Pseudomonas aeruginosa* biofilms at higher concentrations (around 300 µM). The phthiocerol dimycocerosate (PDIM)-deficient *Mycobacterium tuberculosis* H37Ra was more sensitive to the ion complexes of HA compared to other mycobacterial strains. Furthermore, HA10FeCl could increase the susceptibility of *Mycobacterium bovis* BCG to vancomycin. Proteomic profiles showed that the potential targets of HA10FeCl were mainly related to mycobacterial stress adaptation, involving cell wall lipid biosynthesis, drug resistance and tolerance and siderophore metabolism. This study provides new insights regarding the antimycobacterial activities of HA and their complexes, especially about their potential anti-biofilm activities.

## 1. Introduction

Tuberculosis (TB) is a communicable disease caused by the pathogenic bacillus *Mycobacterium tuberculosis*. According to the World Health Organization (WHO) Global Tuberculosis Report 2022, globally, in 2021, an estimated 10.6 million people developed active tuberculosis and 1.58 million died from tuberculosis [1]. In addition, the occurrence of increasing numbers of multi-drug-resistant (MDR) and extensively drug-resistant (XDR) *Mycobacterium tuberculosis* strains is of great concern [1]. The cell wall of *Mycobacterium tuberculosis* contains, from the inside to the outside, an arabinogalactan layer covalently attached to a peptidoglycan layer and mycolic acids covalently attached to the arabinogalactan, in which various unique long carbon chain lipids can be further embedded. The complex outer membrane is further wrapped up by capsular and biofilm extracellular compounds [2]. Some of the long and complex lipids, which are non-covalently attached to the outer membrane and are only present in pathogenic mycobacteria, such as phthiocerol dimycocerosate (PDIM), have been shown to play important roles, not only in virulence, i.e., for camouflage in macrophages, but also in antibiotic intrinsic resistance [3,4]. Furthermore, pathogenic mycobacteria show slow growth, tending to enter a non-replicative state when facing in vivo stress conditions, eventually leading to dormancy. To circumvent this metabolic characteristic, long-term (more than 6 months) and complex (more than four drugs) medications must be administered to patients in order to eradicate the infection. Despite the efficacy of current anti-TB treatments, several factors, such as poor patient compliance, the toxicity of drugs and irrational prescribing practices, increase the risk of drug-resistant strain selection. This leads to even more difficult and expensive treatments [1]. Thus, novel and efficient antituberculosis drugs are urgently needed to fight TB.

Since the first hydroxamic acid (HA) discovery in 1869, HA and its related analogs have been intensively studied. Interestingly, some HA display antimicrobial properties with antibacterial, antiviral or antifungal activities, among others, by chelating metal ions.

For instance, cyclic hydroxamic acid (*N*-hydroxylactam), which is able to chelate iron ions, inhibits the iron-containing lipoxygenase in Gram-negative *P. aeruginosa* [5]. Heterocyclic methylsulfone hydroxamate analogs inhibited the activity of the LpxC enzyme, which is involved in the lipopolysaccharide (LPS) production of Gram-negative bacteria [6]. The metal ions from the first-row transition metals, such as manganese, iron, cobalt, nickel, copper and zinc, are required for these vital processes. It was reported that metal-ion-containing enzymes catalyze around 50% of biochemical reactions in bacteria [7]. Indeed, the maintenance of metal ion homeostasis is a huge challenge for microorganisms, as these ions should be in sufficient amounts to meet cellular demands while staying under the cytotoxicity level. Cells have developed various metallo-regulatory mechanisms for maintaining the necessary homeostasis of metal ions for diverse cellular processes. So, chelating can affect, for instance, metallo-enzymes promoting complex biochemical reactions, metal transporters involved in critical metal ion homeostasis and metal-responsive transcriptional regulators modulating gene expression [8]. On the other side, some metal compounds with low complexation with metals, such as the HA–metal ion derivatives, can act as antimicrobial agents by delivering toxic metal concentrations. In particular, high copper concentrations can affect proteins involved in protein biosynthesis and cellular redox homeostasis [7]. Iron homeostasis is also essential, as high iron concentrations could lead to the production of hydroxyl radicals by interaction with reactive oxygen species [9]. Iron–sulfur proteins, which are involved in various physiological processes, can also be targeted by copper, cobalt or zinc [10]. Competitive or unspecific metal binding to proteins could thus explain the antimicrobial activities of metal ions, including nickel ions, eventually leading to oxidative stress [11]. Indeed, some antibiotics (e.g., bacitracin and bleomycin) require metal ions to perform their functions; thus, metal ions are tightly bound to them, constituting a fundamental part of the drugs and playing an important role in maintaining the proper structure and/or efficiency of their action [12]. Due to the potential values of HA analogues in the field of medicine development, we previously synthesized various HA derivatives, including complexes with varying numbers of carbon atoms (C_2_, C_6_, C_8_, C_10_, C_12_ and C_17_) and their corresponding Fe(II/III), Ni(II), Cu(II) and Zn(II) complexes, and we observed that C_6_ to C_12_ HA derivatives complexed with Fe(II/III) had antibacterial activities against Gram-positive and Gram-negative bacteria, while C_12_ HA had larger-spectrum activities on bacteria and yeast [13]. C_12_ HA compounds complexed or not with Fe(II/III) even showed antimycobacterial activity against *M. smegmatis* [13]. The antimycobacterial activity of HA analogs was previously reported on various mycobacteria, like *M. tuberculosis*, *M. abscessus*, *M. marinum* and *M. smegmatis* [14,15,16]. The complexes of benzohydroxamate associated with transition metallic ions (Cu^2+^ and Co^2+^) could inhibit *M. tuberculosis* growth by interacting with urease in the nitrogen metabolism [14]. *Para*-nitrobenzohydroxamic acid demonstrated a minimum inhibitory concentration (MIC) of 0.71 μM in a glycerol–alanine salt medium or an MIC of 7.79 μM in a 7H12 medium on *M. tuberculosis* [17]. The pentacyanoferrate moiety in Fe(II) coordination hydroxamic complexes benefited the release of HNO from HA, improving pyrazinamide and delamanid efficiency against *M. tuberculosis* [18]. Suberoylanilide hydroxamic acid (SAHA) also had adjunctive potential to enhance the effects of first-line anti-TB drugs (isoniazid and rifampicin) against intracellular *M. tuberculosis* [19].

The aims of this study were to evaluate the potential and selective antimycobacterial and anti-biofilm activities (against *M. tuberculosis*, *M. bovis* BCG, *M. marinum* and *P. aeruginosa*) of 47 HA derivatives, containing various numbers of carbon atoms (C_2_, C_6_, C_8_, C_10_, C_12_ and C_17_) and eventually complexed with Fe(II/III), Ni(II), Cu(II) and Zn(II), to further assess their potential as anti-TB or anti-biofilm drug candidates.

## 2. Material and Methods

### 2.1. Materials

The mycobacterial strains, including *M. bovis* BCG (Pasteur 1173P2), *M. tuberculosis* H37Ra and *M. marinum* (*M* strain), were obtained from Gene Optimal (Shanghai, China). *P. aeruginosa* (LMG 6395) was purchased from the Belgian Coordinated Collection of Microorganisms (BCCM), University of Gent. The 7H9 broth and OADC medium were purchased from BD (BD BBL^TM^, Franklin Lakes, NJ, USA). The Mueller Hinton broth (MHB) was purcHAed from Sigma-Aldrich (Saint Louis, MO, USA). The HA derivatives, composed of various carbon atoms (C_2_, C_6_, C_8_, C_10_, C_12_ and C_17_) and eventually complexed with Fe(II/III), Ni(II), Cu(II) and Zn(II), were prepared as previously described and finally solubilized in DMSO or methanol [13].

### 2.2. Methods

#### 2.2.1. Antimycobacterial Drug Susceptibility Assay

Mycobacterial precultures were grown in 25 cm^2^ flasks at 37 °C without shaking in 7H9 medium (BD) supplemented with 10% (*v*/*v*) OADC (BD) to an OD_600_ of 0.7–0.9. A macrodilution method was performed in 10 mL screw tubes in 7H9 medium (BD) containing 0.05% (*v*/*v*) glycerol and 10% (*v*/*v*) OADC. A total of 500 μL of inoculum diluted in the supplemented 7H9 medium to reach an optical density at 600 nm (OD_600_) of 0.01 was added to 500 μL of serial drug dilutions in the same 7H9 medium. The tubes were placed without shaking at 37 °C. Growth or absence of growth were recorded on the day that the growth of the 100-fold-diluted drug-free inoculum control became visible in order to assess the minimal inhibitory concentration of the drugs (MIC being the lowest drug concentration inhibiting more than 99% of mycobacterial growth) [20]. This experiment was performed three times to check the reproducibility of the MIC determination.

The combined effect of vancomycin and the chemical component was also performed using a microdilution method. The fractional inhibitory concentration index (FICI) was calculated according to the checkerboard method.
FICI = FIC_a_ + FIC_b_ = MIC_ab_/MIC_a_ + MIC_ba_/MIC_b_(1)

MIC_a_ and MIC_b_ represent the MIC of the chemical component or vancomycin alone. MIC_ab_ is the MIC of the chemical component in combination with a fixed vancomycin concentration. MIC_ba_ is the MIC of vancomycin in combination with a fixed chemical component concentration. The fixed concentrations were 4- and 20-fold lower than the MIC_a_ or MIC_b_, respectively. In agreement with the checkerboard method, synergy is reached when the FICI is <0.5 [21]. 

#### 2.2.2. Biofilm Growth Assay

For the mycobacteria, precultures were grown in a 7H9 medium with 10% (*v*/*v*) OADC to an optical density at 600 nm (OD_600_) of 0.8–0.9. A biofilm growth inhibition assay was performed in 6-well or 24-well plates in Sauton’s medium [22,23]. All the compound samples were first diluted in Sauton’s medium to various concentrations. The mycobacterial precultures were inoculated as a 100-fold dilution (corresponding to approximately 9 × 10^5^ CFU/mL inoculum) in the plates, reaching a final culture volume of 4 mL or 2 mL in the 6-well or 24-well plates, respectively. The plates were covered with two layers of Parafilm^®^ “M” and incubated at 37 °C. Biofilm formation was visually assessed after 3–4 weeks of culture.

In contrast to the mycobacterial biofilm inhibition test, the *P. aeruginosa* biofilm inhibition test was performed on pre-formed biofilms [24]. *P. aeruginosa* pre-formed biofilms were obtained on the TSP (solid-phase transfer) covers (Nunc) of the 96-well plates from a 10^6^ CFU/mL starting inoculum incubated for 24 h in MHB. After 24 h, the covers with pre-formed biofilms were immersed and incubated at 37 °C for twelve days in 100 µL of MHB containing compound samples at different concentrations ranging from 20 to 2500 µM. Cetrimide was used as a control (concentrations ranging from 78 ng/mL to 1 µg/mL). Biofilm control wells developed in the presence of 5% (*v*/*v*) DMSO or MeOH were present on each plate. The TSP coverslips containing the previously formed biofilms were used to cover the supports of the plates containing the test compound solutions. These plates were incubated at 37 °C for twelve days [24]. During the incubation period, observation of the plates was performed daily to ensure the presence of the MHB medium. The experiment was performed twice in triplicate for each solution of the tested compounds.

### 2.3. Proteomic Analysis

In order to identify potential targets of the HA complexes, we performed a proteomic analysis. The *M. bovis* BCG (Pasteur 1173P2) strain was treated with HA10FeCl for 7 days, and mycobacteria without HA10FeCl were used as a control. The bacterial pellets were collected by centrifugation at 3000× *g* for 15 min and washed twice with cold PBS. The bacteria were resuspended and lysed in SDT lysis buffer (4% *m*/*v*, SDS, 100 mM Tris-HCl pH 7.6, 1 mM DTT). The extracted proteins were quantified with the BCA Protein Assay Kit (Bio-Rad, Hercules, CA, USA). Protein digestion by trypsin was performed according to a filter-aided sample preparation (FASP) procedure [25]. The digested peptides of each sample were desalted on C18 cartridges (Empore SPE cartridges, Sigma, Saint Louis, MO, USA), concentrated by vacuum centrifugation and reconstituted in 40 µL of 0.1% (*v*/*v*) formic acid.

An LC-MS/MS analysis was performed on a TimsTOF Pro mass spectrometer (Bruker, Billerica, MA, USA) that was coupled to Nanoelute (Bruker Daltonics, Billerica, MA, USA) for 60/120/240 min. The peptides were loaded onto a reverse-phase trap column (Thermo Scientific, Waltham, MA, USA) connected to a C18 reverse-phase analytical column (Thermo Scientific Easy Column, 10 cm long, 75 μm inner diameter, 3 μm resin) in buffer A (0.1% (*v*/*v*) formic acid) and separated with a linear gradient of buffer B (84% (*v*/*v*) acetonitrile and 0.1% (*v*/*v*) formic acid) at a flow rate of 300 nL/min. The mass spectrometer analysis was performed in the positive ion mode. The mass spectrometer collected ion mobility MS spectra over a mass range of *m*/*z* 100–1700 and 1/k0 of 0.6 to 1.6 and then performed 10 cycles of PASEF MS/MS, with a target intensity of 1.5 k and a threshold of 2500. Active exclusion was enabled with a release time of 0.4 min.

The MS raw data were combined and searched using the MaxQuant software (version: 1.5.3.17) for identification and quantitation analyses and the UniProt *Mycobacterium bovis* (strain BCG/Pasteur 1173P2) database (https://www.uniprot.org/, accessed on 25 July 2023). The protein hits with a *p* value of < 0.05 (*t* test) and a fold change of <0.9 or >1.1 were further analyzed.

## 3. Results

### 3.1. HA Compounds Present Antimycobacterial Activities

In order to identify whether the HA derivatives have antimycobacterial activities, we assessed these activities in planktonic and stress conditions with high glycerol concentrations leading to biofilm development. Biofilm conditions are indeed known to better mimic stressful in vivo conditions [26,27,28]. These antimycobacterial activities were assessed using three strains, harboring various characteristics, among others, the presence or absence of PDIM lipids in their cell wall: the H37Ra *M. tuberculosis* strain is PDIM-negative and is a slow-growing strain; the *M. marinum* is PDIM-positive and a slow-growing mycobacterium (however, in planktonic cultures, it grows faster than *M. bovis* BCG or H37Ra *M. tuberculosis* strains); and the *M. bovis* BCG is PDIM-positive and a slow-growing mycobacterium. The drug susceptibility assays performed under planktonic growth conditions eventually allowed for the identification of the minimal inhibitory concentration (MIC), while those performed under biofilm growth development eventually allowed for the identification of the minimal biofilm-formation inhibitory concentration (MBIC).

In planktonic conditions, various HA and complexes displayed antimycobacterial activities, especially on the *M. tuberculosis* H37Ra strain (Table 1, Table 2 and Table 3). Generally, the active compounds had an MIC of ≥200 µM on the three species. An inhibitory activity was less often observed for *M. marinum*, with only HA10 and the HA8FeCl and HA8Fe3 complexes showing antimycobacterial activity. For the *M. bovis* BCG, eight compounds showed inhibitory activity, however, with a relatively high MIC (≥200 µM), except for HA10FeCl, which showed an MIC of 100–200 µM.

Mycobacteria were grown into biofilm under stressful conditions. Biofilm formation was visible from 10 to 12 days for *M. bovis* BCG and *M. tuberculosis* H37Ra and from 15 to 20 days for *M. marinum*. Mature biofilms were obtained in 20–24 days for *M. bovis* BCG, in 28–30 days for H37Ra *M. tuberculosis*, and in 35–40 days for *M. marinum*. The biofilm was thinner for *M. marinum* compared to those of *M. bovis* BCG and H37Ra *M. tuberculosis*. Again, more compounds had biofilm inhibitory activities on the H37Ra *M. tuberculosis* strain. HA10 was active on the three biofilms with a minimal biofilm-formation inhibitory concentration (MBIC) ranging between 100 and 250 µM, depending on the bacteria strain. For most active compounds in the *M. marinum* biofilm, the MBIC was ≥200 µM, except for HA8FeCl, HA10FeCl, HA10Fe and HA8Ni2, which had an MBIC of 62.5 µM, and for HA12Ni2 and HA12Cu2, which had an MBIC of 125 µM. For *M. bovis* BCG, the active compounds had an MBIC of generally ≥100 µM, except for HA10Fe2 (20 µM), HA10FeCl (20–100 µM), HA10Fe3 (20–40 µM), HA10Ni2 (20 µM) and HA10Cu2 (20 µM) (Appendix A) (Table 1, Table 2 and Table 3). For the H37Ra *M. tuberculosis* strain, the active compounds generally had an MBIC of ≥125 µM, except for HA6Ni2, HA8Ni2, HA10FeCl and HA8Fe3 (an MIC of 31.25 µM), HA10Fe3 (an MIC of 31.25–62.5 µM) and HA17Fe3 (an MIC of 62.5 µM) (Table 1, Table 2 and Table 3).

In order to assess whether some of the HA compounds could target PDIM biosynthesis in pathogenic mycobacteria, we investigated the susceptibility of *M. bovis* BCG to vancomycin in the presence of HA10FeCl. Indeed, we previously showed that vancomycin, which is usually used to treat Gram-positive bacteria and is inactive on pathogen mycobacteria, can target those ones when they are lacking PDIM in their cell wall [4]. Drug-targeting the compounds involved in PDIM biosynthesis can thus synergize with vancomycin in the drug susceptibility assay. Interestingly, in the present study, the complex HA10FeCl increased the susceptibility of *M. bovis* BCG to vancomycin by more than 4-fold in the drug susceptibility assay. To investigate whether this growth inhibition results from a synergy between HA10FeCl and vancomycin, the checkerboard method was also used to calculate FICI (Table 4). The MICs of the HA10FeCl complexes dropped from between 46.34 and 92.7 μg/mL to 11.59 μg/mL in the presence of vancomycin (50 μg/mL), and the MIC of vancomycin dropped from 750 µg/mL to 125 µg/mL in the presence of the HA10FeCl complexes (46.35 µg/mL), suggesting that the combination can inhibit *M. bovis* BCG growth in synergy (FICI = 0.292–0.417).

### 3.2. HA10Fe2, HA12Fe2 and HA12FeCl Can Also Reduce Pre-Formed P. aeruginosa Biofilms

Furthermore, in view of the large spectrum of compounds able to inhibit mycobacterial biofilm development, we investigated the anti-biofilm activity of the compounds on a pre-formed *P. aeruginosa* biofilm, a well-known biofilm difficult to eradicate. Most of the compounds were totally inactive in this assay, such as HA10Ni2, HA10Cu2, HA8Cu2, CuCl_2_, NiCl_2_ and FeCl_2_, even at 2.5 mM. Interestingly, the iron complexes of HA10Fe2, HA12Fe2 and HA12FeCl inhibited *P. aeruginosa* biofilm formation with an MBIC of 625 µM, 312.5 µM and 312.5 µM, respectively.

### 3.3. Proteomic Profile of the HA10FeCl-Treated Bacilli

In order to better understand the mode of action of the drug candidate, we also compared the proteomics of the HA10FeCl-treated *M. bovis* BCG cells with untreated cells (Figure 1 and Figure 2). A total of 41 proteins were screened and identified with significant differences (*p* < 0.05) by *t* test and with a fold change (FC) of >1.1 and <0.9, including 32 up-regulated and 9 down-regulated proteins in the HA10FeCl-treated bacilli compared with the control without treatment (Appendix A).

Generally, hydrolase proteins encoded by the genes *pstC2*, *bglS*, *BCG_1059c*, *BCG_0364*, *moaC3*, *gltD*, *lipG*, *BCG_2973*, *BCG_0099*, *truA*, *BCG_1935c*, *mltG*, *topA* and *pks12* were up-regulated. The possible membrane proteins encoded by the genes *BCG_1478*, *BCG_3932*, *BCG_3545c*, *mmpS4* and *BCG_1127c* were also up-regulated in the HA10FeCl-treated bacilli. The phosphate transport system permease protein encoded by the gene *pstC2* with an FC of 2.19 and rubredoxin encoded by *BCG_3279c* with an FC of 2.19 were also up-regulated. The proteins encoded by the genes *crgA*, *BCG_3932*, *BCG_1708*, *BCG_1384c*, *BCG_0352*, *BCG_2826*, *BCG_3492c*, *gltD*, *lipG*, *BCG_2813c*, *BCG_0099*, *BCG_1268*, *truA*, *mutT2*, *BCG_1935c*, *mltG*, *topA*, *BCG_3927*, *BCG_1127c* and *rnpA_2* were also up-regulated in HA-treated *M. bovis* BCG. In addition, the proteins encoded by *pks12* (*BCG_2067c*) and *BCG_2973* involved in PDIM biosynthesis and the protein mmpS4 involved in siderophore export were also up-regulated in HA-treated *M. bovis* BCG. By contrast, the proteins encoded by the genes *thiG*, *BCG_2384c* and *BCG_2177c* were down-regulated in the HA10FeCl-treated *M. bovis* BCG.

Furthermore, we also observed some up-regulated proteins, encoded by *BCG_1100c*, *narK2*, *BCG_0259c* and *BCG_3092* in HA10FeCl-treated *M. bovis* BCG, and down-regulated proteins, encoded by genes *PE15* and *mmpL9a* (*BCG_2361*) in the control, that were not identified from the proteomic profiles (Appendix A).

## 4. Discussion

In the present study, antimycobacterial activities and anti-biofilm activities were assessed using three mycobacterial strains (H37Ra *M. tuberculosis*, *M. bovis* BCG and *M. marinum*) and *P. aeruginosa*. Globally, the obtained results showed that HA antimycobacterial activity depended on the carbon chain length and metal ions in the complexes. HA with a C_10_ and C_12_ carbon chain (HA10 and HA12, respectively) displayed higher activity than the C_2_ and C_6_ carbon chain HA (HA2 and HA6, respectively). Among the complexes, those with a C_8_, C_10_ and C_12_ carbon chain, for instance, HA10FeCl, HA12FeCl, HA8Fe3 and HA10Fe3, showed interesting antimycobacterial activities with an MIC of 125 µM and an MBIC of 31.25–62.5 µM on *M. tuberculosis*, with an MBIC similar to the Ni complexes HA6Ni2 and HA8Ni2. The iron complexes HA10FeCl, HA10Fe2 and HA10Fe3 displayed higher inhibitory activity on *M. bovis* BCG biofilm with an MBIC of 20–100 µM compared to the other iron complexes. By contrast, these iron complexes showed less activity on *M. bovis* BCG growth in planktonic conditions. Indeed, the iron complexes’ inhibitory effect on mycobacterial biofilm formation is globally better than in planktonic growth conditions.

In this study, our compounds were evaluated to target three mycobacterial strains: *M. bovis* BCG (PDIM^+^/PGL^+^), *M. marinum* (PDIM^+^/PGL^+^) and *M. tuberculosis* H37Ra (PDIM^−^). As expected from a PDIM-negative strain, the *M. tuberculosis* H37Ra strain was more susceptible to a larger panel of compounds compared to the other two mycobacterial strains. This was probably due to impaired cell wall impermeability. Mycobacterial cell wall lipids, like trehalose monomycolate and dimycolate (TMM, TDM), PDIM, sulpholipid-1 (SL-1), diacyl trehalose (DAT) and pentacyl trehalose (PAT), play an important role in mycobacterial pathogenesis and impermeability. Consequently, proteins involved in lipid biosynthesis or transport are considered potential anti-TB drug targets. In *M. tuberculosis*, the membrane protein MmpL and the MmpS family play important roles not only in the transport of important cell wall lipids across the mycobacterial membrane [29] but also in drug efflux (MmpL5 and MmpL7), siderophore export (MmpL4/MmpS4 and MmpL5/MmpS5) and heme uptake (MmpL3 and MmpL11) [30,31]. The up-regulation of some of these membrane proteins in the HA10FeCl-treated *M. bovis* BCG compared to untreated bacteria suggests that these HA derivatives could affect PDIM biosynthesis and siderophore transport. This is in agreement with the synergy observed between H10FeCl and vancomycin to inhibit *M. bovis* BCG growth. As iron–sulfur proteins [3Fe-4S] and [4Fe-4S] ferredoxins, [1Fe-0S] rubredoxin, the ortholog RubB from *M. tuberculosis*, which is involved in redox homeostasis, also participated in the electron transfer, notably during the oxidative stress process [32,33,34]. Cytochrome P450 (CYP) enzymes catalyze primary and secondary metabolite production and participate in environmental toxin and drug oxidation. For catalysis, CYPs require electrons typically supplied by redox proteins, such as ferredoxins [34]. Our results also suggest that HA10FeCl participates in the ferredoxin and flavodoxin balance in *M. bovis* BCG by decreasing CYP124 production and up-regulating rubredoxin for adaptation to the stress H10FeCl treatment condition.

As is well known, the virulent mycobacterial cell wall lipids PDIM and PGL were considered potential targets for antituberculosis drug development. The phenolphthiocerol (phthiocerol) moiety biosynthesis of PGL (PDIM) requires several type I polyketide synthases (PKS) encoded by *ppsA*-*E* and *pks15*/*1*, among others, like Pks12, Pks18, TesA, DrrB, Pks6, PpgK, ClpS and Mmpl7 [35,36,37,38,39,40]. The expression of *pks1* also correlates with other gene expressions, like the expression of *fadD22*, *Rv2949c*, *lppX*, *fadD29*, *pks6* and *pks12* [41]. The largest open-reading frame (*pks12*) in the genome of *M. tuberculosis* H37Rv encodes probable polyketide synthase needed to produce fatty acids, probably involved in the synthesis of phthiocerol, the diol required for DIM synthesis [37]. Indeed, the phthiocerol/phenolphthiocerol dimycocerosates methyltransferase encoded by the gene BCG_2973 (Rv2952 in *M. tuberculosis*) could catalyze the reduction of phthiodiolone and phenolphthiodiolone to yield phthiocerol and phenolphthicerol in *M. tuberculosis* [42]. The proteins from the proteomic profiles, encoded by the genes *pks12* and BCG_2973, were significantly up-regulated, further indicating that HA10FeCl could affect the virulent PDIM/PGL biosynthesis. 

Additionally, the phosphate-specific transport operon *pstS3-pstC2-pstA1*, regulated by the two-component regulatory system (2CRS) SenX3-RegX3 in *M. tuberculosis* during phosphate depletion and nutrient starvation [43] and potentially required for isoniazid’s high susceptibility [43,44], is up-regulated (at least PstC2) in HA10FeCl-treated bacilli. It will therefore be interesting to evaluate in the future if HA10FeCl could increase the susceptibility to isoniazid and if their combination could be of use to fight against the isoniazid-resistant *M. tuberculosis*.

Furthermore, the membrane protein PE15, which is a PE/PPE protein and essential for efficient Ca^2+^ uptake [45], and the efflux pomp MmpL9a (BCG_2361) were not at all detected in HA10FeCl-treated *M. bovis* BCG, suggesting either their lower expression compared with the control or their loss due to a defected outer membrane, i.e., by a reduced amount of PDIM.

## 5. Conclusions

In summary, out of HA and its Fe(II/III), Ni(II), Cu(II) and Zn(II) complexes evaluated for their antimycobacterial activity, the most promising growth inhibitors were some Fe(III), Cu(II) and Zn(II) complexes (HA10FeCl, HA10Fe3, HA8Fe3, HA10Zn2 and HA12Cu2), which exhibited the highest antibacterial activity against pathogenic mycobacteria. Fe complexes showed stronger inhibitory effects on the PDIM-deficient *M. tuberculosis* strain than on the other two strains in stressful conditions. Their mechanisms of action still need to be further investigated, among others, by AFM-IR. Indeed, the proteomic data and synergistic activity of the H10FeCl compound suggest that it could affect the integrity of the mycobacterial cell wall.

## Figures and Tables

**Figure 1 microorganisms-11-02611-f001:**
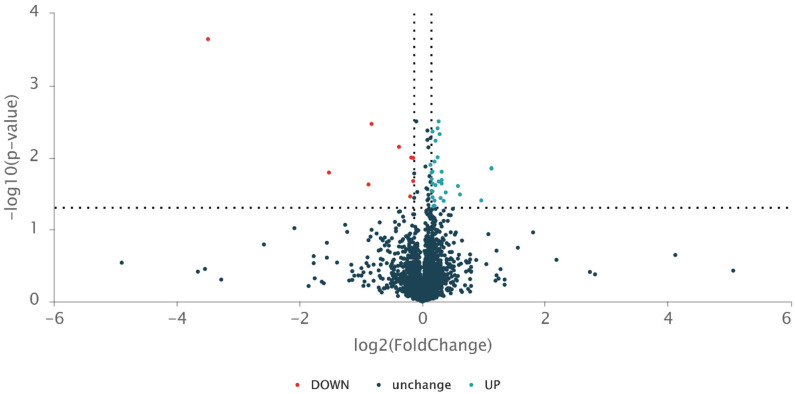
Volcano plot of proteomic data of HA10FeCl-treated *M. bovis* BCG versus untreated *M. bovis* BCG control. The dotted line parallel to the x axis indicates a y value of 1.301 (i.e., −log_10_ 0.05). The two dotted lines parallel to the y axis indicate x values of −0.152 (i.e., log_2_ 0.9) and 0.138 (i.e., log_2_ 1.1), respectively. Three independent samples for each group were subjected to a proteomic analysis. Fold changes relative to the HA10FeCl-treated bacilli are shown in Appendix A.

**Figure 2 microorganisms-11-02611-f002:**
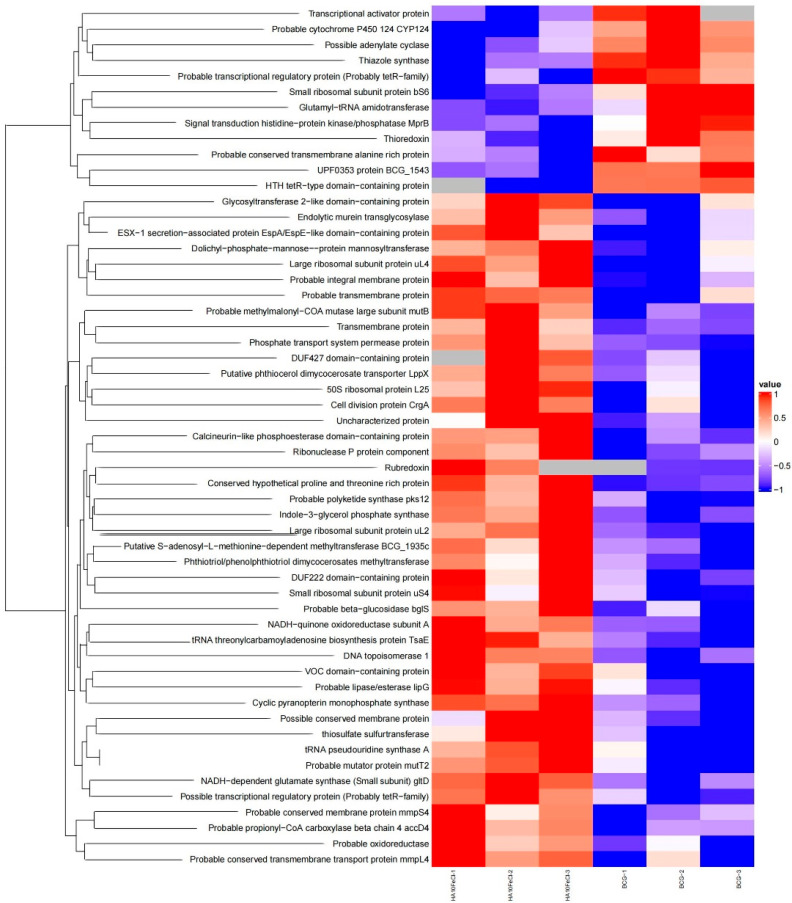
Proteomic analysis of HA10FeCl-treated *M. bovis* BCG versus untreated *M. bovis* BCG control. The heatmap shows differentially expressed proteins in HA10FeCl-treated *M. bovis* BCG and the control. Colors changing from blue to red indicate increased levels of proteins. The gray color indicates proteins not detected in proteomic profiles. Three independent samples for each group were analyzed for proteomic analysis.

**Table 1 microorganisms-11-02611-t001:** MIC and MBIC values (µM) of HA–iron complexes against *M. bovis* BCG, *M. marinum* and *M. tuberculosis*.

Compound	*M. bovis* BCG		*M. marinum*		*M. tuberculosis* H37Ra	
	MIC	MBIC	MIC	MBIC	MIC	MBIC
	µM	µM	µM	µM	µM	µM
HA2FeCl	>500	500	>500	250	>500	250
HA6FeCl	>500	300	>500	≥250	500	125
HA8FeCl	≥500	100	200	62.5	500	250
HA10FeCl	100–200	20–100	>200	62.5	125	31.25
HA12FeCl	>200	100	>200	500	125	125
HA17FeCl	>500	>500	>500	500	250	>500
HA2Fe2	>500	>500	>500	>500	>500	>500
HA6Fe2	>500	100–500	>500	500	500	250
HA8Fe2	>500	100–200	>500	500	500	250
HA10Fe2	>200	20	>200	>500	250	>500
HA12Fe2	>500	100	>500	>500	500	500
HA17Fe2	250	>500	>500	500	125	250
HA2Fe3	250–500	300–500	>500	250	>500	125
HA6Fe3	500	100–500	>500	250	≥500	62.5–125
HA8Fe3	500	100	250–500	≥125	125	31.25
HA10Fe3	≥200	20–40	>200	62.5	125	31.25–62.5
HA12Fe3	≥200	100	>200	200–500	125–250	125
HA17Fe3	>200	>500	>500	200–500	125	62.5

HA = hydroxamic acid, HAnFe2 = iron(II) complexes, HAnFeCl = iron(III) complexes with n = 2, 6, 8, 10, 12 and 17 (carbon chains).

**Table 2 microorganisms-11-02611-t002:** MIC and MBIC values (µM) of zinc, nickel(II) and copper(II) complexes against *M. bovis* BCG, *M. marinum* and *M. tuberculosis*.

Compound	*M. bovis* BCG		*M. marinum*		*M. tuberculosis* H37Ra	
	MIC	MBIC	MIC	MBIC	MIC	MBIC
	µM	µM	µM	µM	µM	µM
HA2Zn2	>500	>500	>500	500	>500	250
HA6Zn2	>500	200	>500	500	125	125
HA8Zn2	>500	100–200	>500	500	250	250
HA10Zn2	250	100	250	250–500	250	250
HA12Zn2	>500	100	>500	250	250	125
HA17Zn2	>500	>500	>500	500	>500	250
HA2Ni2	>500	300–500	>500	>500	>500	62.5–125
HA6Ni2	>500	200	>500	500	>500	31.25
HA8Ni2	>500	100	>500	62.5	>500	31.25
HA10Ni2	>200	20	>200	500	500	125
HA12Ni2	>500	100	>500	125	>500	250–500
HA17Ni2	>500	>500	>500	250	>500	250
HA2Cu2	250–500	300	>500	≥250	≥500	250
HA6Cu2	≥500	100	>500	500	>500	500
HA8Cu2	>200	100–200	>200	250	>500	250
HA10Cu2	>200	20	>200	250	250	250
HA12Cu2	>500	100	>500	125	>500	250
HA17Cu2	>500	≥500	>500	≥500	>500	250

HAZn2 = zinc complexes, HAnNi2 = nickel(II) complexes, HAnCu2 = copper(II) complexes with n = 2, 6, 8, 10, 12 and 17 (carbon chains).

**Table 3 microorganisms-11-02611-t003:** MIC and MBIC values (µM) of HA complexes and iron(II), iron(III), nickel(II), copper(II) and zinc(II) chloride against *M. bovis* BCG, *M. marinum* and *M. tuberculosis*.

Compound	*M. bovis* BCG		*M. marinum*		*M. tuberculosis* H37Ra	
	MIC	MBIC	MIC	MBIC	MIC	MBIC
	µM	µM	µM	µM	µM	µM
HA2	>500	>500	>500	500	>500	250
HA6	>500	~100	>500	500	500	125
HA8	>500	100	>500	500	500	125
HA10	250–500	100	250–500	250	>500	125
HA12	>500	100–500	>500	>500	125	250
HA17	>500	>500	>500	≥500	≥500	>500
FeCl_2_	>500	>500	>500	500	>500	>500
FeCl_3_	>500	>500	>500	500	>500	>500
NiCl_2_	>200	>500	500	≥500	>500	250
CuCl_2_	>200	200	>200	250	>500	500
ZnCl_2_	>500	>500	>500	250	>500	125–250

HA complexes with n = 2, 6, 8, 10, 12 and 17 (carbon chains).

**Table 4 microorganisms-11-02611-t004:** Vancomycin susceptibility assay obtained in a macrodilution series and analyzed by the checkerboard method for *M. bovis* BCG strain.

Compounds	MIC (μg/mL)/FICI
Vancomycin	750/-
HA10FeCl	48.18–96.37/-
HA10FeCl (46.35 μg/mL) + Vancomycin	125/0.176
Vancomycin (50 μg/mL) + HA10FeCl	11.59/0.125–0.25

-: means without calculation (it is not possible to obtain FICI in the presence of only one drug in the experiment according to Equation (1)).

## Data Availability

The data presented in this study are available in this article and Appendix A.

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
