# Peer review of "Antimycobacterial Activities of Hydroxamic Acids and Their Iron(II/III), Nickel(II), Copper(II) and Zinc(II) Complexes"

_microorganisms, 2023, doi:10.3390/microorganisms11102611_

Round 1

Reviewer 1 Report

The manuscript with the title "Antimycobacterial activities of hydroxamic acids and their iron(II/III), nickel(II), copper(II) and zinc(II) complexes" is devoted to the urgent problem of finding new complex antimycobacterial pharmaceuticals to prevent the growth of biofilms antibiofilm activities of mycobacteria. According to the authors' concept, hydroxamic acids with iron(II/III), nickel(II), copper(II) and zinc(II) complexes should increase the antibiotic activity of hydroxamic acids. The number of compounds used by the authors is very large and the authors have done a significant amount of work. It is also important to note the proteomic studies conducted by the authors, which allow us to describe the mechanisms of antimicrobial activity of one of the complexes. In general, I can say that the manuscript corresponds to the main aims and scopes of the journal "Microorganisms".

I have several comments on the manuscript:

1. In the abstract, the authors should describe the specific results of the study.

2. In the introduction, it is worth adding works that provide data on the influence of the metals studied by the authors on the activity of mycobacteria. It is necessary to explain the choice of these metals in the work.

3. The authors should describe the data in Figure 1 more clearly and simply. It is generally difficult for an unprepared reader to understand

4. The authors should supplement the conclusion with more practical conclusions.

Minor editing of English language required

Reviewer 2 Report

The authors have showcased an innovative work, but crucial methodological details are missing along with essential experiments necessary in these kinds of antimicrobial studies. The authors need to extend their work for further consideration. The study is based on in vitro data and there has to be sufficient data to support their claims.

1.      The authors want to see the synergistic antibacterial effect of the metal component with HA. In this case, the introduction must have references or previous research works related to the usage of these metals in the field of bacterial control.

2.      The authors probably forgot to add a crucial methodology portion in this manuscript related to the synthesis of the HA complexes with metals or they failed to add the sources of these complexes which makes the manuscript incomplete.

3.       The bacterial growth inhibition zones under different treatment conditions need to be shown in terms of images for the selected best candidates as a proof statement. The quantified numbers for MIC and MBIC are not enough evidence in these kinds of studies. Supporting data has to be there as proof.

4.      Proteomics data alone is not sufficient to support the results. Further proof is needed. Metals usually attach to the bacterial membranes and increase ROS levels along with damage to bacterial DNA inside. Please show bacterial DNA fragmentation data to support the claims for bactericidal effect by the HA metal complexes.

5.      Supplementary Figure 1 is not provided to the reviewer which has been mentioned by the authors in the contextual part.

The English language can only be considered once the manuscript contents are sufficient for a full-scale review.

Round 2

Reviewer 1 Report

The authors significantly revised the manuscript, taking into account all my comments. I hope that in this form the manuscript can be accepted for publication.

Reviewer 2 Report

The authors have fairly improved the manuscript based on suggestions provided and it is ready to be accepted although the authors must proceed with the promised future work.